# High Numbers and Densities of PD1^+^ T-Follicular Helper Cells in Triple-Negative Breast Cancer Draining Lymph Nodes Are Associated with Lower Survival

**DOI:** 10.3390/ijms21175948

**Published:** 2020-08-19

**Authors:** Peter Bronsert, Anna von Schoenfeld, Jose Villacorta Hidalgo, Stefan Kraft, Jens Pfeiffer, Thalia Erbes, Martin Werner, Maximilian Seidl

**Affiliations:** 1Institute of Surgical Pathology, Faculty of Medicine, Medical Center, University of Freiburg, 79106 Freiburg, Germany; peter.bronsert@uniklinik-freiburg.de (P.B.); a.willemer@hotmail.de (A.v.S.); JoseVillacorta.Hidalgo@miltenyi.com (J.V.H.); martin.werner@uniklinik-freiburg.de (M.W.); 2Tumorbank, Comprehensive Cancer Center Freiburg, Medical Center, Faculty of Medicine, University of Freiburg, 79106 Freiburg, Germany; 3German Cancer Consortium (DKTK) and Cancer Research Center (DKFZ), 69120 Heidelberg, Germany; Thalia.erbes@uniklinik-freiburg.de; 4Faculty of Medicine, University of Freiburg, 79106 Freiburg, Germany; pfeiffer@hnoamtheater.de; 5Center of Dermatopathology, Freiburg, 79106 Freiburg, Germany; stefkraft@hotmail.com; 6Department of Oto-Rhino-Laryngology, Medical Center, University of Freiburg, 79106 Freiburg, Germany; 7Department of Obstetrics and Gynecology, Medical Center, University of Freiburg, 79106 Freiburg, Germany; 8Institute of Pathology, Heinrich Heine University and University Hospital of Duesseldorf, 40225 Duesseldorf, Germany

**Keywords:** tumor draining lymph node, breast cancer, PD1^+^ T-follicular helper cells, germinal center

## Abstract

Breast cancer tumor draining lymph nodes (TDLNs) display distinct morphologic changes depending on the breast cancer subtype. For triple-negative breast cancers (TNBC), draining LNs display a higher amount of secondary lymphoid follicles, which can be regarded as a surrogate marker for an activated humoral immune response. In the present study, we focus on PD1^+^ T-follicular helper cells (Tfh) in TDLNs of TNBC, since PD1^+^ Tfh are drivers of the germinal center (GC) reaction. We quantified PD1^+^ Tfh in 22 sentinel LNs with 853 GCs and interfollicular areas from 19 patients with TNBC by morphometry from digitalized immunostained tissue sections. Overall survival was significantly worse for patients with a higher number and area density of PD1^+^ Tfh within GCs of TDLNs. Further, we performed T-cell receptor gamma chain (TRG) analysis from microdissected tissue in the primary tumor and TDLNs. Eleven patients showed the same TRG clones in the tumor and the LN. Five patients shared the same TRG clones in the tumor and the GCs. In two patients, those clones were highly enriched inside the GCs. Enrichment of identical TRG clones at the tumor site vs. the TDLN was associated with improved overall survival. TDLNs are important relays of cancer immunity and enable surrogate approaches to predict the outcome of TNBC itself.

## 1. Introduction

Breast cancer prognosis has improved dramatically in recent years due to targeted therapeutic options. Nevertheless, triple-negative breast cancer (TNBC) is associated with poor patient overall survival. Tumor immunology is presently in the focus of therapeutic cancer research, not only for breast cancer. Immune checkpoint blockade shows promising results for TNBC [1,2] and has become a promising treatment for TNBC [3]. In general, PD-1/PD-L1 checkpoint inhibitors “revive” T-cells (”exhausted” cytotoxic T-cells) within the tumor microenvironment by blocking the PD-1/PD-L1 interaction, which tumors use to inhibit a T-cell mediated immune response. As predictive surrogate markers for immune checkpoint inhibitor therapy, T-cell infiltration, tumor PD-L1 expression, and tumor mutational load are used [4,5]. Interestingly, besides local intra- and peritumoral factors, systemic surrogate markers, such as PD-L1 expressing eosinophils and lymphocytes within the peripheral blood may also be useful [6,7].

Studies analyzing PD-1/PD-L1 expression in tumor-draining lymph nodes (TDLN) have remained scarce. In animal models, TDLNs have been shown to be key regulators in antitumor immune response and control the magnitude of therapeutic efficacy of PD-1 and PD-L1 blocking [8]. TDLNs represent the first sites of tumor metastasis and are resected in most diseases. Furthermore, apart from the tumor environment, TDLN also represent the first site of tumor-induced immune modulation.

Considering the prognostic and predictive role of the lymph drainage via sentinel LNs and the standardization of surgical regimes according to national applied guidelines, breast cancer represents a proof-of-concept entity for the investigation of cancer-based immune modulating effects.

The interactions within and between the different lymph node compartments play an important role in the initiation and regulation of immune responses [9]. Via T-cell priming and suppression, TDLNs have been described balancing both the induction and the suppression of tumor immunity [10,11]. Tumor antigens are transported by CD103^+^ dendritic cells, initiating T-cell priming [12,13]. Furthermore, TDLNs of TNBC display higher densities of germinal centers (GCs), indicative of higher humoral activity [14]. The immune cell composition and spatial distribution in TDLNs may have a predictive value for immune-based interventions. We, therefore, studied the distribution of PD1^+^ T-follicular helper cells (Tfh) in the GCs, the distribution of CD8^+^ cytotoxic T-cells together with morphometrically assessed LN compartments and clonal distribution of T-cells. We demonstrate a potential predictive role of TDLNs in immunotherapy for breast cancer. In the clinical setting, our study may bear relevance with regard to the necessity for sentinel LN dissection.

## 2. Results

All correlation coefficients, corresponding *p*-values, and numbers are provided in the supplemental table.

### 2.1. Study Cohort

From 2008–2015, 19 patients with TNBC with at minimum one tumor draining LN, all treated in an adjuvant setting, were included into this pilot study. Patients’ ages ranged from 35 to 79 years (mean 66.32 years, standard deviation (SD) 12.82 years). In sum, 22 analyzed tumor draining LNs displayed 853 GCs, which were analyzed for area, roundness, and cellular composition of Bcl6^+^ PD1^+^ Tfh. CD8^+^ cytotoxic T-cells were quantified in GCs and interfollicular areas. As expected, the tumor size positively correlated with the T-stage and the N-category with the number of affected LNs (Appendix A). HER2/neu IHC score ranged from 0 to 2+ (mean 0.53; SD 0.7). Subsequent chromogenic in situ hybridization revealed no HER2/neu positive tumor (Table 1 displays the overall description of the study cohort’s patient characteristics). Out of 19 patients, 15 patients are deceased. Event-specific overall survival ranged from 5–72 months, with a mean 26.73 months and an SD of 22.59 months. Four patients were still alive after a 72 month follow-up.

### 2.2. Higher Numbers and Higher Densities of PD1^+^ Tfh Are Associated with Shorter Survival

To better understand where humoral immune reactions are taking place—especially in draining LNs of TNBC–PD1^+^—Tfh were quantified and correlated with clinical and histological data. The cellular density was calculated by number of PD1^+^ cells over the area of the corresponding GC. Using the Pearson’s correlation coefficient, high mean numbers of PD1^+^ Tfh were associated with shorter event-specific survival (*cc* = −0.614; *p* = 0.015; *n* = 15), which was also true for higher mean densities of PD1^+^ Tfh (*cc* = −0.604; *p* = 0.017; *n* = 15), which is visualized exemplarily in Figure 1A,B. Subsequently, we performed a Kaplan Meier survival plot, based on a mean split separation of the groups with either higher or lower mean PD1^+^ Tfh. The results are displayed in Figure 1B,C, further supporting the finding of an adverse outcome in TN patients with higher PD1^+^ Tfh densities. The descriptive statistics of the cellular measurements are summarized in Table 2.

### 2.3. Higher Numbers of CD8^+^ GC Cells Are Associated with Elevated GC Amount, Enlarged GCs, and a Higher Tumor Grade

Based on the PD1^+^ Tfh findings suggestive for a tumor associated humoral activation of the TDLN, we speculated that CD8^+^ cells as possible cytotoxic counterparts might be correlated with improved patients’ survival. We did not find any significant correlation in this direction. However, higher mean numbers of CD8^+^ GC cells significantly correlate with higher GC numbers (*cc* = 0.497; *p* = 0.03; *n* = 19), larger mean, but not median, GC areas (*cc* = 0.624; *p* = 0.004; *n* = 19), and higher mean and median CD8^+^ GC cell area densities (*cc* = 0.565 and 0.54; *p* = 0.012 and 0.017; *n* = 19 and 19). Concerning the clinical parameters, a weaker positive correlation for higher mean CD8^+^ GC cell numbers and the tumor grading (*cc* = 0.458; *p* = 0.049; *n* = 19) was detectable. The findings are visualized in Figure 1E–I. Looking for the pT category, only median numbers of CD8^+^ GC cells (*cc* = −0.468; *p* = 0.043; *n* = 18) and interfollicular area density of CD8^+^ cells (*cc* = −0.486; *p* = 0.041; *n* = 18) displayed an inverse correlation. The descriptive statistics of the cellular measurements are summarized in Table 2, of the GC measurements are summarized in Table 3.

### 2.4. Identical TRG/T-Cell Clones Are Present in the Tumor, the Draining LN, and the GC in the Majority of Patients. Higher T-cell Clone Tumor/LN Ratios Are Associated with Improved Patients’ Overall Survival

Next, we investigated whether the above-mentioned effect of PD1^+^ Tfh numbers in GCs on the patients’ overall survival might be driven by several identical, probably expanded T-cell clones in the tumors and the corresponding LNs and GCs. From 14 out of 19 patients enough LN tissue was available to perform TRG clonality analysis of LN and matched tumor samples. Within this subgroup, eleven patients are deceased. Eleven out of 14 patients showed identical T-cell clones in the tumors and the draining LNs (clone numbers range 0–391; mean 69.9; standard deviation 109.6). The number of identical T-cell clones between the tumor and the draining LN was higher in patients with LN metastases (*cc* = 0.584; *p* = 0.028; n = 14). Looking into the percentages of the identical T-cell clones within the tumor (tumor infiltrating T-lymphocytes, TILs) and the corresponding LN revealed that the identical T-cell clones being a minor fraction in the TIL and the LN T-cell population (Figure 2(BI,BII)). However, if the identical clones occupy a larger fraction of the TILs compared to the fraction in the corresponding LN T-cell population, the patients display an improved overall survival (Figure 2A).

Considering GCs, 10 of 14 patients contained enough GCs for TRG clonality analyses. Thereof, 5 of 10 patients showed the same T-cell clones within the tumors and GCs (clone numbers range 0–107; mean 14.2; standard deviation 33.1; findings are summarized in Table 4).

Next, we asked which percentage of identical clones can be found in the tumor infiltrating lymphocytes, the draining LN, and the GCs, respectively. Regarding the identical clones between tumor and LN in the TIL population, higher percentages correlated with higher circularity of the GCs (*cc* = -0.594; *p* = 0.025; *n* = 14 for mean value and *cc* = −0.621; *p* = 0.018; *n* = 14 for median value of non-circularity). Furthermore, measuring the percentage of the TRG reads allowed us to estimate if certain clones are enriched in different compartments (Figure 2B). The enrichment of the same clones in the population of TILs is comparable to the one in LN derived lymphocytes ( 2(BI,BII)). Identical T-cell clones between the tumor and GC represent only a very small TIL fraction (Figure 2(BIII)), which is also the case vice-versa. However, two patients show a high percentage in the GC of clones identified in the tumor samples (Figure 2(BIV)), speaking for a high enrichment inside the GC compartment.

## 3. Discussion

In our previous work, we found that TN breast cancer is associated with higher amounts of GCs in the tumor draining LNs, especially in contrast to the luminal subtypes [14]. TN breast cancers are also known to harbor higher amounts of PD1^+^ tumor infiltrating “exhausted” cytotoxic T-cells, indicative for a tumor microenvironment of an increased PD1 signaling [15]. On the humoral axis, PD1 signaling represents the key for the function of PD1^+^ Tfh, which are drivers of the GC reaction [16,17]. We, therefore, hypothesized that PD1^+^ Tfh cells play a role in the immune reaction of TN breast cancers, leading to this pilot study. We did not find the nodal status being significantly correlated with patients’ overall survival (*cc* = −0.277; *p* = 0.318; *n* = 15), keeping in mind the relatively low number of patients. We saw that the sizes, shapes, and cellular composition of the examined draining LNs displayed a spectrum, wherein higher numbers and densities of PD1^+^ Tfh cells are associated with an adverse outcome. Considering this, the number of PD1^+^ Tfh could be a further marker to predict the potential response of the patient towards an immune checkpoint inhibitor therapy. Beyond this possible clinical impact, the mechanistic story behind this finding is yet untold. Under physiological conditions, higher numbers PD1^+^ Tfh cells are associated with higher numbers of antigen specific B-cells [18,19,20], which is important for a successful humoral immune response. Hence, shaping a TDLN towards a humoral immune response would lead to a lower survival rate in TN breast cancer patients. An approach to identify a possible source of this immune response is to directly isolate plasma cells from tumor draining LNs and check whether they secrete antibodies binding to tumor antigens, e.g., by indirect immunofluorescence of tumor sections incubated with supernatant from cultured plasma cells [21]. However, the role of antibodies against tumor antigens needs further clarification and could be complemented by techniques to identify foreign peptides [22,23] as antibodies against tumor antigens are insufficient to carry an antitumoral immune response [24].

Interestingly, we did not find associations between important mediators of cellular immunity, namely CD8^+^ T cells, and survival. However, higher numbers of CD8^+^ GC cells are associated with larger GCs, a finding we already knew from patients with common variable immunodeficiency [25] and which recently could be linked to an exhaustion phenotype with, e.g., coexpression of PD1 of those follicular CD8^+^ cells [26]. This could explain their lacking impact on survival in our cohort. Moreover, their positive correlation with tumor grading is suggestive of at least some tumor-dependent modulation of CD8^+^ GC cells, yet a causative relationship is not proven by this finding.

Identical T-cell clones between tumor and draining LNs are a common finding in our cohort, but these identical clones are the minor population in the entirety of TILs or the LN T-lymphocytes, based on the number of reads. However, patients showed an improved survival when the population of identical T-cell clones is enriched in the tumor (Figure 2B) vs. the LN, which is highly suggestive for an expansion of anti-tumoral T-cells at the effector site. Two patients showed a strong enrichment of tumor T-cell clones inside the GC, which makes them possible candidates of mediating a tumor-antigen-dependent humoral response. It should be noted, though, that the majority of patients did not show this enrichment of tumor T-cell clones inside the GCs. It is, therefore, not clear whether tumor-antigen-specific T-cells shape the draining lymph node or are just modulated by, e.g., soluble factors, which modulate T-cells with different antigen specificity in the same way. Future experiments should, therefore, address the functional phenotype of the T-cells, as well as their clonality. Furthermore, dissecting the different Tfh subtypes (Tfh1, Tfh2, Tfh17) would lead to a better understanding of which co-stimulatory signal is behind the immune reaction and could, therefore, be a candidate to be therapeutically antagonized, e.g., in an antitumoral vaccination approach.

## 4. Materials and Methods

### 4.1. Patients and Cohort

All patients in the present study have given written informed consent before inclusion. Ethics approval was obtained by local authorities of the Ethics Committee of the University Medical Center Freiburg (Ref: 10011/16; 25 April 2016). Surgery for primary breast cancer and corresponding LNs was performed at the Department of Obstetrics and Gynecology at the University Medical Center Freiburg. Only patients histopathologically diagnosed for triple-negative breast cancer were included. In order to prevent the interference of neo-adjuvant therapeutic interventions with LN morphology and function, only adjuvant-treated patients were included. Clinico-pathological data comprised UICC-, WHO- and R- classification, tumor grading (according to Elston and Ellis [27]), and immunohistological subtype (according to Goldhirsch et al. [28]) as well as patients’ overall survival (which equals patients’ disease-free survival in this cohort). All tissue specimens and corresponding data were distributed by the Tumorbank of the Comprehensive Cancer Center Freiburg and pseudonymized before distribution. Nineteen patients with tumors and 22 draining lymph node samples were included.

### 4.2. Multiple Immunohistochemical Staining

The immunohistochemical stainings were performed on sections of formalin fixed, paraffin-embedded lymph nodes. In short, 2 μm-thick sections from each sentinel lymph node were transferred to a 39 °C water bath before mounting on slides. Slides were dried at 58 °C overnight. After drying, all sections were de-paraffinized in a descending alcohol series and then transferred to wash buffer. CD8 (ready to use antibody, clone C8/144B, Mouse Anti-Human, monoclonal, Dako Envision Flex REF IR623, Dako Agilent, Glostrup, Denmark) was stained in red after heat mediated epitope retrieval for two minutes at citric buffer pH6. PD-1 (ready-to-use antibody, Goat Anti-Human polyclonal, R&D Systems AF 1086, Minneapolis, MN, USA) was stained in blue and BCL6 (ready-to-use antibody, Pg-B6, Mouse Anti-Human, monoclonal, Dako Envision Flex IR625) in brown after heat mediated epitope for five minutes in citric buffer at pH6. Formalin-fixed and paraffin-embedded tonsil and lymph node specimens, derived from the archive of the Institute for Surgical Pathology at the University Medical Center Freiburg, served as positive/negative control. Internal control comprises the characteristic distribution pattern of cells in the applied stainings. In detail, PD1^+^ Tfh are represented in the light zone of the GC, CD8^+^ cells are predominantly allocated at the interfollicular region, BCL6^+^ cells are predominantly allocated inside the GC (Figure 1A,B). After performing the triple staining, the sections were counterstained with hematoxylin for 45 s followed by a washing step for five minutes under running, deionized water. Finally, all slides were mounted with aqueous mounting medium (Aquatex, Merck Millipore, Darmstadt, Germany) and covered with coverslips (Langenbrinck, Emmendingen, Germany).

### 4.3. Slide Scanning and Evaluation of LNs

Multistained LNs were digitized using a Mirax Scan Panoramic Scanner (20× objective). All scans were evaluated using the Definiens Tissue Studio for Immunohistochemistry analysis software (Definiens AG, Germany, Versions 4.3.1–4.4.1). 

Definiens Tissue Studio was trained for LN architecture (whole LN, GC, and interfollicular zone) and for the subcellular localization of the applied immunohistochemical staining (CD8 and PD1 membranous, BCL6 nuclear) using the provided workflow “Solution” for image analysis. From the trained parameters, an analysis algorithm is created. For the respective coloring within the CD8/BCL6/PD1, two Solution workflows were generated. In both, the GCs, the interfollicular region and the surrounding connective tissue were defined as regions of interest (ROIs). The areas were calculated from all ROIs. The first Solution workflow was applied to calculate the roundness of the GCs and to determine the number of PD1^+^ cells inside the GC. To identify only PD1 expressing Tfh, only PD1 and BCL6 co-expressing cells were analyzed. The second Solution workflow enumerated the CD8^+^ cells within the GCs and the interfollicular region. Of note, automated analysis was visually reviewed during the software training by experienced pathologists, and randomly selected GCs and interfollicular zones were manually counted for Solution workflow verification. GC ROIs with an area of at least 5000 μm^2^ were included in the evaluations.

### 4.4. Laser-Capture Microdissection, DNA Extraction, and NGS Analysis of TCR Gamma Clonality

In order to investigate the presence of mutual T-cell clones, we investigate the CDR3 region of the TRGamma gene locus (TRG) by NGS in tumor and draining lymph nodes. We selected the TRG because of its ubiquitous rearrangement in T–cells. Additionally, the TRG CDR3 region represents a sensitive T-cell marker for tumor and lymph node tissue specimens where a dissection into separate cell populations before DNA/RNA extraction was not possible [29]. DNA was isolated from 5 µm FFPE slides of tumors, LNs and—if possible—from 5–10 pooled GCs, isolated by laser capture microdissection (LCMD) after mounting on membrane coated slides (Zeiss LaserMicrobeamMicrodissection System with LaserPressureCatapulting, RoboCut Software and Membrane Slide NF 1.0 PEN, respectively; Carl Zeiss Microscopy GmbH, Göttingen). Before LCMD, individual GCs were marked and separated into areas of 30.000 μm^2^. Marked areas were extracted by pulse laser beam and catapulted into the lid of Adhesive Cap Tubes (Carl Zeiss Microscopy GmbH, Göttingen). Each tube contained LCMD tissue from two serial sections of the same patient derived from different GCs. After LCMD of up to ten GCs, the whole LN was macrodissected and transferred to a separate tube. After dissection, tissue specimens were incubated with 25 μL 1:20 PCR buffer (Roche Applied Biosystems, Mannheim; 1:10 diluted in AccuGENE Molecular Biology Water, Lonza, Basel, Switzerland) and treated with proteinase K (Proteinase K 20 mg/mL, Genaxxon Bioscience, Ulm) at 58 °C overnight. Proteinase K was inactivated at 95 °C for 10 min via the thermocycler. All tubes were centrifuged for one minute. TRG sequencing and analysis was performed using the LymphoTrack^®^ Dx TRG Assay Kit according to the manufacturer’s guidelines, including the software. The resulting sequences were screened for same sequences in tumors vs. LNs, tumors vs. GCs, and LNs vs. GCs, respectively, using a software-based approach with string operators, quantified based on their percentage of reads and visualized (Python 3.7, Anaconda distribution; Pandas, Numpy and Matplotlib package; software scripts can be provided upon request by the authors). The length of 130 bases covered the CDR3 region and was used to identify the same sequences throughout the different isolates. Duplicates were eliminated afterwards. The numbers of matching clones in the tumor, their draining LNs and their GCs were quantified. Furthermore, the percentage of reads of the respective T-cell clones were summarized to quantify the clonal density in the different compartments (tumor vs. LN vs. GC).

### 4.5. Statistical Analysis

The data collected by Definiens Tissue Studio were processed using Microsoft Excel 2016. Kolmogorov–Smirnov adaption test (www.novustat.com, 26.06.2018) was performed for testing normal distribution. According to the distribution, scale parameters were expressed as median, mean, and range, respectively. Ordinal and nominal variables were expressed as absolute numbers. Further statistical testing was performed with a two-sided significance level below *p* = 0.05 by Pearson test for correlations coefficients, using a Python-based solution (Python 3.7, Pandas and Numpy package; script can be provided upon request by the authors). Logrank test, and Cox proportional hazards regression for survival were performed as well (Python 3.7, Pandas, Matplotlib and Lifelines package; jupyter notebook can be provided upon request by the authors).

## 5. Conclusions

As higher amounts of PD1^+^ Tfh are correlated with an adverse outcome in TN breast cancer patients in our pilot study, further studies are needed to identify the antigens, co-stimulatory factors and tumor-derived factors driving this humoral immune response. Therefore, the sentinel LN as a key relay of the local immune response necessarily must be taken into account for an improved understanding of the tumor immunology.

## Figures and Tables

**Figure 1 ijms-21-05948-f001:**
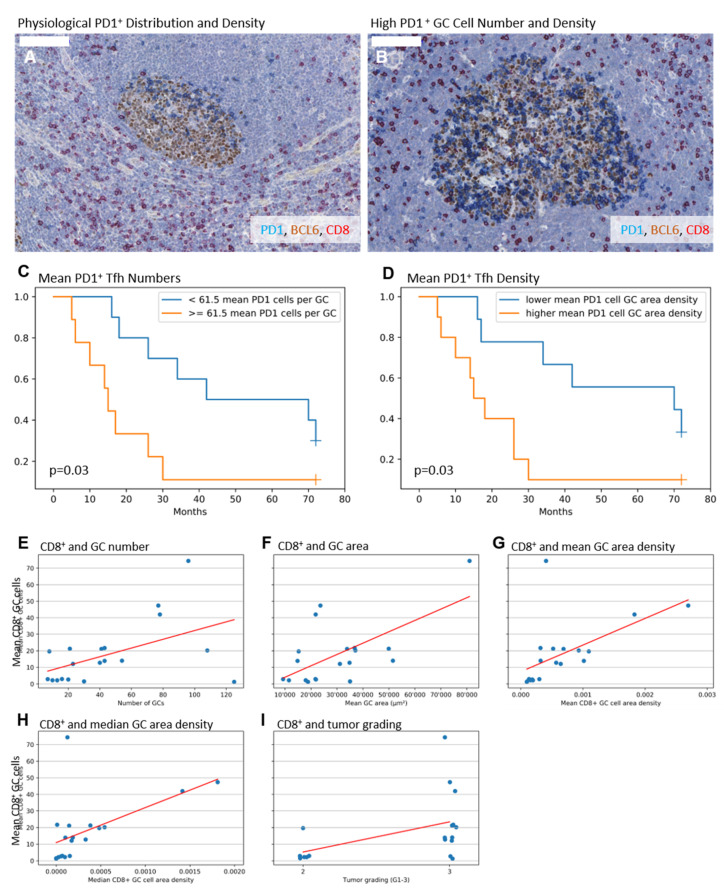
(**A,B**) Exemplary immunohistochemical triple stainings showing BCL6^+^ germinal centers (GCs, brown), PD1^+^ T-follicular helper cells (Tfh, blue), and CD8^+^ cytotoxic T-cells (red); 100µm indicated by white bar in the upper left, respectively. (**A**) Exemplary GC with polarized PD1^+^ Tfh distribution, lower PD1^+^ Tfh number and lower area density. (**B**) Exemplary GC with non-polarized PD1^+^ Tfh distribution, higher PD1^+^ Tfh number and higher area density. (**C**–**E**) Separation of groups by mean split. (**C**) Mean PD1^+^ Tfh number ≥ 61.5 cells per GC is associated with lower survival (*p* = 0.03, logrank test, 15 deaths for 72 months follow-up observation time, 1 patient still alive in the PD1 high group, 3 patients still alive in the PD1 low group after 72 months follow-up). (**D**) Mean density ≥ 0.002475 PD1^+^ Tfh per mm^2^ GC area is associated with lower survival (*p* = 0.03, logrank test, 15 deaths for 72 months follow-up observation time, 1 patient still alive in the PD1 high group, 3 patients still alive in the PD1 low group after 72 months follow-up). (**E**–**I**) Association between mean numbers of CD8^+^ GC cells, morphometrical, and clinical findings. (**E**) Higher mean numbers of CD8^+^ GC cells are associated with higher GC numbers, higher mean GC areas (**F**), higher mean and median CD8^+^ GC cell density (**G**,**H**), and higher tumor grade (**I**).

**Figure 2 ijms-21-05948-f002:**
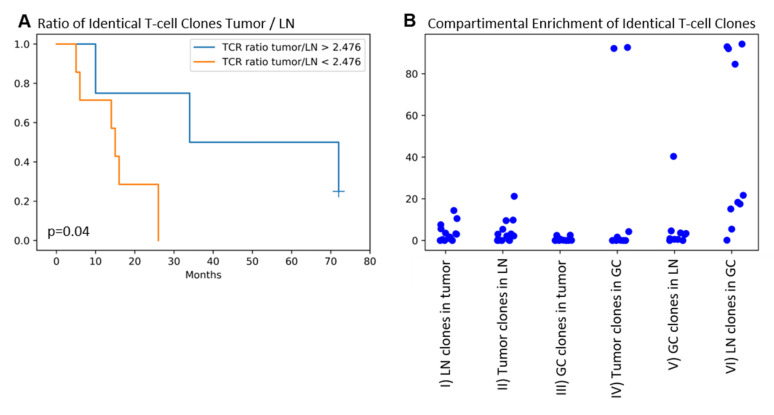
(**A**) Tumor/lymph node ratio of identical T-cell clones > 2.476 is associated with improved survival (*p* = 0.04, logrank test, 11 deaths for 72 months follow-up observation time, 1 patient still alive in the high ratio group after 72 months follow-up; ratio was calculated as (sum of % TRG reads tumor)/(sum of % TRG reads lymph node) of identical TRG sequences from tumors and matched lymph nodes). (**B**) Displays the percentage (%) of reads (TRG sequencing) of identical clones in different compartments to see whether clones are expanded: (**I**) identical clones between draining lymph node (LN) and tumor: % reads in the tumor sample; (**II**) identical clones between tumor and draining LN: % reads in the LN sample; (**III**) identical clones between tumor and germinal center (GC): % reads in the tumor sample; (**IV**) identical clones between tumor and germinal GC: % reads in the GC sample, showing two patients with strongly enriched clones inside the GCs; (**V**) identical clones between lymph node and GC: % reads in the LN sample; (**VI**) identical clones between lymph node and GC: % reads in the GC sample.

**Table 1 ijms-21-05948-t001:** Descriptive statistic of patients’ characteristics.

	*n* Patients	Mean	Std	Min	25%	50%	75%	Max
**Age**	19	66.32	12.82	35.00	55.50	72.00	75.50	79.00
**Grading (G1-3)**	19	2.68	0.48	2.00	2.00	3.00	3.00	3.00
**pT**	19	1.53	0.51	1.00	1.00	2.00	2.00	2.00
**pN**	19	0.37	0.76	0.00	0.00	0.00	0.50	3.00
**Number of LN Metastases**	19	0.74	2.28	0.00	0.00	0.00	0.50	10.00
**Total Number of LNs**	19	6.21	8.48	1.00	1.50	3.00	5.00	31.00
**Lymphangiosis (1 = yes, 0 = no)**	19	0.16	0.37	0.00	0.00	0.00	0.00	1.00
**HER2/neu IHC Score**	19	0.53	0.70	0.00	0.00	0.00	1.00	2.00
**MIB-1/Ki67 Proliferation Index**	15	56.13	22.59	17.00	40.00	50.00	75.00	90.00
**Event-Specific Overall Survival in Months**	15	26.73	20.68	5.00	14.50	18.00	32.00	72.00
**Overall Survival in Months**	19	36.26	26.31	5.00	15.50	26.00	71.00	72.00

LN: lymph node.

**Table 2 ijms-21-05948-t002:** Descriptive statistics of cellular measurements.

	*n* Patients	Mean	Std	Min	25%	50%	75%	Max
**CD8^+^ Interfollicular Density (cells per mm^2^)**	18	3228.61	1305.26	1267.61	2314.34	2746.34	4211.88	5714.43
**Interfollicular Area (mm^2^)**	18	49.62	22.83	15.17	32.06	45.51	66.35	97.87
**Mean Number of CD8^+^ Cells per GC**	19	17.67	19.00	1.24	2.75	13.91	21.18	74.38
**Median Number of CD8^+^ Cells per GC**	19	5.47	6.76	0.00	1.00	2.50	7.50	28.00
**Mean GC Density of CD8^+^ Cells (cells per µm^2^)**	19	641.97	664.70	97.10	189.75	406.79	810.49	2694.89
**Median GC Density of CD8^+^ Cells (cells per µm^2^)**	19	321.35	486.22	0.00	61.85	144.99	356.46	1810.47
**Mean Number of PD1^+^ Tfh per GC**	19	61.53	45.42	0.58	23.76	61.17	88.84	158.52
**Median Number of PD1^+^ Tfh per GC**	19	21.84	24.30	0.00	0.00	11.50	38.50	65.50
**Mean GC Density of PD1^+^ Tfh (cells per µm^2^)**	19	2475.33	1928.32	70.60	812.83	2539.88	4354.01	5598.64
**Median GC Density of PD1^+^ Tfh (cells per µm)**	19	1380.91	1749.04	0.00	0.00	849.68	2294.69	6089.25

GC: germinal center; Tfh: T-follicular helper cell.

**Table 3 ijms-21-05948-t003:** Descriptive statistics of GC measurements.

	*n* Patients	Mean	Std	Min	25%	50%	75%	Max
**Number of GCs**	19	44.89	35.77	7.00	18.00	40.00	65.50	125.00
**Sum of GC Areas in mm**	19	1.56	1.82	0.07	0.39	1.05	1.76	7.78
**Ratio: Sum of GC Areas/Whole LN area**	19	0.02	0.02	0.00	0.00	0.01	0.02	0.08
**Mean GC Area in µm^2^**	19	29907.58	17232.53	9303.41	18429.66	23661.85	35912.72	80989.81
**Median GC Area in µm^2^**	19	16521.21	6728.29	7755.20	11012.91	15450.05	21316.81	29980.66
**Mean Circularity Ratio (1 circular, >1 non-circular)**	19	1.24	0.12	1.08	1.15	1.20	1.31	1.54
**Median Circularity Ratio (1 circular, >1 non-circular)**	19	1.17	0.09	1.06	1.09	1.13	1.22	1.43

**Table 4 ijms-21-05948-t004:** Descriptive statistics of T-cell receptor gamma (TRG) sequences of the CDR3 region measurements in the tumor, the draining LNs, and the GCs of these LNs.

	*n* Patients	Mean	Std	Min	25%	50%	75%	Max
**Identical TRG Sequences Between Tumor and LN: Sum of % Reads in Tumor**	14	3.67	4.42	0.00	0.43	2.34	5.08	14.37
**Identical TRG Sequences Between Tumor and LN: Sum of % Reads in LN**	14	4.10	5.95	0.00	0.12	2.16	4.82	21.22
**Identical TRG Sequences Between Tumor and GC: Sum of % Reads in Tumor**	10	0.59	1.03	0.00	0.00	0.04	0.56	2.54
**Identical TRG Sequences Between Tumor and GC: Sum of % Reads in GC**	10	19.07	38.65	0.00	0.00	0.04	3.60	92.62
**Identical TRG Sequences Between LN and GC: Sum of % Reads LN**	10	5.67	12.30	0.00	0.52	1.86	3.56	40.34
**Identical TRG Sequences Between LN and GC: Sum of % Reads GC**	10	44.19	40.79	0.12	15.71	19.98	90.13	94.24
**Number of Identical TRG Sequences Between Tumor and LN**	14	69.93	109.55	0.00	4.50	18.50	108.25	391.00
**Identical TRG sequences: Ratio of Sum % Reads Tumor/Sum % Reads LN**	11	2.48	2.81	0.11	0.58	1.38	3.51	9.01
**Number of identical TRG Sequences Between Tumor and GC**	10	14.20	33.07	0.00	0.00	1.50	7.75	107.00
**Identical TRG Sequences: Ratio of Sum % Reads Tumor/Sum % Reads GC**	5	0.43	0.72	0.02	0.03	0.03	0.42	1.68
**Number of Identical TRG Sequences Between LN and GC**	10	24.80	26.97	2.00	4.00	11.50	47.25	71.00
**Identical TRG Sequences Between LN and GC: Ratio of Sum % Reads LN/Sum % Reads GC**	10	0.33	0.82	0.00	0.01	0.06	0.14	2.67

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
