# Peer review of "High Numbers and Densities of PD1+ T-Follicular Helper Cells in Triple-Negative Breast Cancer Draining Lymph Nodes Are Associated with Lower Survival"

_ijms, 2020, doi:10.3390/ijms21175948_

Round 1
Reviewer 1 Report
High number and densities of PD1+ T-follicular helper cells in triple negative breast cancer draining lymphnode are associated with lower survival
In this pilot study with 19 TNBC patients the authors describe interesting associations between morphometric parameters in draining lymphnodes and patients survival, highlighting the importance of sentinel LNs to understand the tumor immune response. Overall the study is well conducted, but results section should be improved for a better clarity.
Specific (minor) comments:
- It is not clear the reference to the figure and/or tables. All tables should be mentioned in the text and clearly explained.
- PD1+ should be better indicated as PD1+
- add TRG in the abbreviation section
Reviewer 2 Report
In this manuscript, the authors studied the association of PD1+ T-follicular helper (PD1+Tfh) cells in tumor draining lymph nodes (TDLNs) with prognosis in triple negative breast cancer patients. These results have shown that higher numbers and density of PD1+Tfh are associated with short survival of patients. The results are important, but the number of patients is very small. This manuscript was poor written with numerous errors, the tables and figures were not properly referred in the results. The results described in 2.1.3 should be shown in the figure. The rationale for each experiment should be more clearly described. Some of the tables can be arranged differently by switching columns to rows. Each table can be on the same page instead of 2 pages.
Round 2
Reviewer 2 Report
The authors have addressed my concerns in the revised manuscript.
Minor comments:
- There are some errors in the manuscript, please carefully proof read.
- The font size in figure 1E - 1I is too small.